# SPLID: Self-Imitation Policy Learning through Iterative Distillation

## Abstract

Goal-Conditioned continuous control tasks remain challenging due to the sparse reward signals. To address this issue, many relabelling methods like Hindsight Experience Replay have been developed and bring significant improvement. Though relabelling methods provide an alternative to an expert demonstration, the majority of the relabelled data are not optimal. If we can improve the quality of the relabelled data, the sample efficiency, as well as the agent performance, should be improved. To this end, we propose a novel meta-algorithm Self-Imitation Policy Learning through Iterative Distillation (SPLID) which relies on the concept of $\delta$-distilled policy to iteratively level up the quality of the target data and agent mimics from the relabeled target data. Under certain assumptions, we show that SPLID has good theoretical properties of performance improvement and local convergence guarantee. Specifically, in the deterministic environment, we develop a practical implementation of SPLID, which imposes $\delta$-distilled policy by discriminating First Hit Time (FHT). Experiments show that SPLID outperforms previous Goal-Conditioned RL methods with a substantial margin.

## 1 Introduction

Goal-Conditioned Sparse Reward (GCSR) tasks are one of the most challenging tasks in reinforcement learning. In these tasks, the goal is combined with the current state as the input of a policy, and only when the agent successfully achieves the desired goal will the agent be assigned with a nontrivial reward, making the task extremely challenging for random-exploration-based algorithms. In many cases, the GCSR task is closely related to the Multi-Goal task, where the goal is not fixed and can be set anywhere in the state space. In order to solve a GCSR task, an agent has to learn a generalizable solution that can be applied to a variety of similar tasks. For example, robotic object grasping is such a GCSR task: the target object can be placed at any position on the table, henceforth, the robot needs to control its arm to get closer to the position of the object and then grasps it. The objective of learning such a policy is to find a feasible path from the current state to the goal (Tamar et al., 2016). The work of Plappert et al. (2018) provides a series of these Goal-Conditioned tasks in robotics control.

In previous works, reward shaping (Ng et al., 1999), hierarchical reinforcement learning (Dietterich, 2000; Barto & Mahadevan, 2003), curriculum learning (Bengio et al., 2009), and learning from demonstrations (Schaal, 1997; Atkeson & Schaal, 1997; Argall et al., 2009; Hester et al., 2018; Nair et al., 2018) is proposed to tackle the challenges of learning through sparse rewards. All of those approaches heavily rely on prior knowledge or expert demonstrations for a given task. The work of Hindsight Experience Replay (HER) (Kaelbling, 1993; Andrychowicz et al., 2017) is proposed to relabel failed trajectories and assigns hindsight credits as complementary to the primal sparse rewards, to address the difficulty in exploration through the way of Temporal Difference (TD) learning. Differently, the recent work of GCSL (Ghosh et al., 2019b) proposes to solve the GCSR tasks, extending HER through iterative supervised learning, i.e., there are no value networks that are optimized based on TD learning. However, in both HER and GCSL, there is no guarantee on the quality of the relabeled data, nor there is a guarantee on the policy improvement based on those relabeled data. On the other hand, the work of PCHID (Sun et al., 2019) proposes an intuitive approach for GCSR tasks in a supervised learning manner, but the sample efficiency is limited by the explicit curriculum setting.

In this work, we aim at improving sample efficiency in GCSR tasks, by improving the quality of relabeled data. Notice that GCSL performs behavior cloning on *all* the successful experience after relabeled without considering improving the quality of the relabeled data, where *quality* can be measured by the timesteps taken to achieve the goal or other means. Since the quality of the expert demonstration that behavior cloning relies on determines the performance and the sample efficiency of the training, it is natural to ask if we can also improve the quality in the target of the behavior cloning. To this end, we propose a meta-algorithm Evolutionary Policy Distillation (SPLID), motivated by the theoretical insight, and instantiate it with a practical algorithm in the GCSR tasks to further improve learning efficiency.

We summarize our contributions as follows:

- To evaluate the quality of the policy in the Multi-Goal RL, we introduce the concept of $\delta$-distilled policy. Motivated by leveling up the quality of the target policy of the behavior cloning, we propose a novel meta-algorithm called SPLID, which iteratively distills the target policy through pursuing $\delta$-distilled policy and conducting self-imitation learning.

- From the theoretical aspect, we prove that under certain assumptions, SPLID has a monotonic performance improvement guarantee and a local convergence guarantee.

- We specialize such meta-algorithm on a practical implementation in the GCSR tasks with the deterministic environment, where SPLID searches $\delta$-distilled policy by distinguishing First Hit Time (FHT) and enhances exploration by parameter noise.

- We demonstrate the proposed practical implementations on the goal-oriented tasks benchmark and show our method SPLID can work in isolation to solve GCSR tasks with improved sample efficiency than previous methods involving HER, PCHID, GCSL, and ES.

## 2 PRELIMINARIES

**Multi-Goal Markov Decision Process.** We first consider a Markov Decision Process (MDP) denoted by a tuple $\mathcal{M}$ containing $(\mathcal{S}, \mathcal{A}, r, \mathcal{T}, \rho(s_0), \mathcal{G}, \gamma)$, where $\mathcal{S}$ is a finite discrete state space, $\mathcal{A}$ denotes the finite discrete action space , $r(s, g) : \mathcal{S} \times \mathcal{G} \mapsto \mathbb{R}$ is the reward function. In this paper we focus on a special case where the reward function is binary, i.e., tasks with sparse reward. Without loss of generalization, we also assume that reward is bounded in $[0, 1]$. $\gamma \in (0, 1)$ is the discount factor. $\mathcal{T} : \mathcal{S} \times \mathcal{A} \mapsto \Delta(\mathcal{S})$ is the transition kenrel and $\mathcal{T}(s_{t+1} \mid s_t, a_t)$ represents the probability of the state transition from $s_t$ to $s_{t+1}$ given the action $a_t$. Here we denote by $\Delta(\mathcal{X})$ distribution set defined on any measurable set $\mathcal{X}$. $\rho(s_0)$ is the start state distribution. $\mathcal{G}$ is the finite goal space and we assume the goal obeys the fixed distribution $p_\mathcal{G}(\cdot) \in \Delta(\mathcal{G})$ at the beginning of each episode. Policy of the agent takes the form $\pi(a \mid s, g) : \mathcal{S} \times \mathcal{G} \mapsto \Delta(\mathcal{S})$ and $\pi$ induces the state distribution $\rho_\pi^t(\cdot \mid g) = \mathbb{P}(s_t = \cdot \mid \pi, g)$. We denote $\rho_\pi(\cdot \mid g) = \frac{1}{1-\gamma} \sum_{t=1}^\infty \gamma^t \rho_\pi^t(\cdot)$ as the discounted average state distribution induced by $\pi$. For convenience, we assume that the environment transits to the null state with recieving reward 1 each time step once the agent reaches the goal. For the introduction of nerual network, we parameterize the policy $\pi$ by $\pi_\theta$, where parameter lies in a certain region $\theta \in \Theta$. We denote that $\Pi_\Theta = \{\pi_\theta : \theta \in \Theta\}$ the parameterized policy class and assume that optimal policy falls into it, i.e.,$\pi^\star \in \Pi_\Theta$.

**Universal Value Function Approximator** The Universal Value Function Approximator (UVFA) Schaul et al. (2015) extends the state space of *Deep Q-Networks (DQN)* Mnih et al. (2015) to include goal state as part of the input, which is useful in the setting where there are multiple goals to achieve. Moreover, Schaul et al. (2015) shows that in the UVFA setting, the learned policy can generalize to previously unseen state-goal pairs. Specifically, let $\mathcal{S}_{(S,G)} = \mathcal{S} \times \mathcal{G}$ denote the *extended state space* of $\mathcal{M}$ where $\mathcal{G}$ is a finite goal space. Normally, a representation mapping $m(\cdot) : \mathcal{S} \to \mathcal{G}$ is assumed to be known in such multi-goal RL frameworks (Plappert et al., 2018).

**Hindsight Experience Replay.** Learning with sparse rewards in RL problems remains challenging due to the difficulty to reach the reward through random explorations. Hindsight Experience Replay (HER) proposes to relabel the failed rollouts as successful ones (Andrychowicz et al., 2017) as a method to deal with the goal-oriented sparse reward problem. The agent in HER receives a reward when reaching either the original goal or the relabeled goal in each episode by storing both original transition pairs $((s_t, g), a_t, (s_{t+1}, g), r)$ and relabeled transitions $((s_t, g'), a_t, (s_{t+1}, g'), r')$ in the

replay buffer, where $g'$ is a certain hindsight goal that will be visited in the following steps, i.e., there exist $k$ such that $g' = m(s_{t+k})$, and $r'$ is the hindsight reward, which will be 1 when the hindsight goal is achieved.

# 3 SELF-IMITATION POLICY LEARNING THROUGH ITERATIVE DISTILLATION

## 3.1 THEORETICAL FRAMEWORK

First, we consider a multi-goal MDP with infinite horizon, where the discounted goal-specified return $V(\pi, g)$ takes the form

$$V(\pi, g) = \sum_{t=0}^{\infty} \mathbb{E}_{s_t \sim \rho_\pi^t(\cdot|g)}[\gamma^t r(s_t, g)],$$

and $\rho_\pi^t$ is the visitation measure induced by $\pi$. Then the target of the agent is to maximize the discounted return $J(\pi)$, i.e., $J(\pi) = \mathbb{E}_{g \sim p_\mathcal{G}} V(\pi, g)$. This problem can be extremely hard when the goal specified reward $r(\cdot, g)$ is sparse or binary. To address this problem with sample efficiency, we are motivated by behavior cloning that mimics a policy from an oracle in a supervised learning manner, e.g. human experts. To implement such a supervised learning method, one issue faced is the access to the oracle data, which is not available in our problem setting, and the quality of the oracle data also determines the performance of the mimicked policy. Although HER (Rauber et al., 2017) proposes to relabel failure trajectory to successful one with a different goal and GCSL (Ghosh et al., 2019b) applies HER in the behavior cloning method, it is inadequate for GCSL to achieve sample efficiency. Because it considers all the policy reaching the goal as the target in the supervised learning and the quality of the target policy it mimics may be a burden in improving the sample efficiency. To this end, besides the combination of HER and behavior cloning as GCSL do, we desire to improve the target policy dynamically, that is assured by an oracle to be qualified to *lead* the present policy. Such a process corresponds to the policy distillation, where it is oriented with an evolutionary target policy selected by an oracle. We will introduce how we implement such oracle-guided policy distillation by the paradigm of behavior cloning to boost sample efficiency from the theoretical perspective. Similar to previous work (Ghosh et al., 2019b; Xu et al., 2019), we only study the discrete case from the perspective of theory. To utilize the relabeling strategy (Rauber et al., 2017) to address GCSR, we first analyze the formal definition of relabeling strategy.

**Definition 1** (Relabeling Strategy). *We define the relabeling strategy of policy $\pi$ with initial goal $g$ in the $t$-timestep as $\eta_\pi^t(g' \mid g) : \mathcal{G} \times \mathcal{G} \times \mathbb{N} \times \Pi_\Theta \mapsto \Delta(\mathcal{G})$.*

By definition 1, relabeling strategy is a time varying distribution, reflecting the relabeling preference w.r.t. the time step in one trajectory. We remark that in HER, such relabelling distribution matches the average state visitation distribution induced by the model policy $\eta_\pi^t(\cdot \mid g) \propto \sum_{i=t}^{\infty} \rho_\pi(\cdot \mid g)$, and in GCSL it matches the last step visitation distribution (finite horizon $T$) i.e., $\eta_\pi^t(\cdot \mid g) = \rho_\pi^T(\cdot \mid g)$, for all $g \in \mathcal{G}$ and $\pi \in \Pi_\Theta$. The next step is to formally evaluate the quality of the target policy that acts as the target of behavior cloning. We introduce the definition of $\delta$-distilled policy as follows.

**Definition 2** ($\delta$-Distilled Policy). *We define the $\delta$-distilled policy $\tilde{\pi} \in \Pi_\Theta$ of a policy $\pi \in \Pi_\Theta$, if the following condition is satisfied:*

$$G(\tilde{\pi}\|\pi) := \sum_{g' \in \mathcal{G}} \mathbb{E}_{g \sim \rho_{\tilde{\pi}}(m^{-1}(\cdot)|g')} \left[ p_\mathcal{G}(g) \cdot (V(\tilde{\pi}, g) - V(\pi, g)) \right] \geq \delta,$$

*where $\delta$ is a nonnegative constant and $\rho_{\tilde{\pi}}(m^{-1}(\cdot) \mid g')$ is defined as $\sum_{\{s \in \mathcal{S} : m(s) = \cdot\}} \rho_{\tilde{\pi}}(m(s) \mid g')$.*

Intuitively, if $\tilde{\pi}$ is the $\delta$-distilled policy of $\pi$, it means that $\tilde{\pi}$ performs nearly strictly better than $\pi$ for the goals that $\tilde{\pi}$ reached. Here the initial goal distribution $p_\mathcal{G}(g)$ acts as the weight in the weighted sum, reflecting the prior favor of such performance advantage. We remark that $\delta$-distilled policy does not directly give the relationship of the return $J(\tilde{\pi})$ and $J(\pi)$ because of the covirate shift in the goal distribution. Nevertheless, such $\delta$-distilled policy is easier to search than the policy with higher return, since we only need to focus on the goals reached by the $\delta$-distilled policy itself rather than the entire goal space $\mathcal{G}$.

Before we move on, we first need to avoid the meaningless case when $\delta$ keeps zero, which is achieved by the following assumption.

**Assumption 1** (Avoid $\delta$ Keeping Zero ). *We assume that*

$$supp(p_{\mathcal{G}}(\cdot)) \cap supp(\rho_{\pi}(m^{-1}(\cdot) \mid g)) \neq \Phi,$$

*for all goal $g \in \mathcal{G}$ and policy $\pi \in \Pi_{\Theta}$, where $\Phi$ is the empty set.*

Assumption 1 excludes the environments where the goal-oriented regions under $p_{\mathcal{G}}$ are blocked away from the agents. This assumption can be easily satisfied by a wide variety of tasks in RL and multi-goal RL, e.g. goal-reaching tasks.

Based on the assumption and previous intuitive insights, we present a meta-algorithm in Algorithm 1 to solve GCSR tasks iteratively, which is named **S**elf-Imitation **P**olicy **L**earning through **I**terative **D**istillation (SPLID) and involves three main stages for each episode.

**Stage 1: Exploration** To search for such $\delta$-distilled policy, we need produce new behavior policy $\tilde{\pi}^{(k)}$ from old model policy $\pi_{\theta^{(k-1)}}$ with the enhancement of exploration. To ensure robustness, we control such exploration span by constrainting the total variation between $\tilde{\pi}^{(k)}$ and $\pi_{\theta^{(k-1)}}$ as

$$\sup_{g \in \mathcal{G}, s \in \mathcal{S}} D_{TV}(\tilde{\pi}^{(k)}(\cdot \mid s, g) \| \pi_{\theta^{(k-1)}}(\cdot \mid s, g)) \leq \epsilon, \tag{1}$$

where $\epsilon$ is an absolute constant.

**Stage 2: Distill the Target** We owe an oracle to judge whether this new behavior policy $\tilde{\pi}^{(k)}$ is our desired $\delta$-distilled policy. If it is, then we mix $\tilde{\pi}^{(k)}$ with old target policy $\pi^{(k-1)}$ with a certain proportion, otherwise we skip this stage without changing target policy, i.e., $\pi^{(k)} = \pi^{(k-1)}$. The mixture process satisfies that

$$\rho_{\pi^{(k)}} = (1 - \alpha_k)\rho_{\pi^{(k-1)}} + \alpha_k \rho_{\tilde{\pi}^{(k)}}, \tag{2}$$

where $\alpha_k \in (0, 1)$ is the mixture proportion at the $k$-th episode. Such a mixture process is also used for ensure the smoothness of the update.

**Stage 3: Behavior Cloning** To update parameter $\theta^{(k)}$, we conduct behavior cloning towards the $\pi^{(k)}$ after relabeling, where the relabeled strategy is $\eta$. If we define the surrogate function $F_{\eta}(\pi_{\theta} \| \bar{\pi})$ as

$$F_{\eta}(\pi_{\theta} \| \bar{\pi}) := \mathbb{E}_{g \sim p_{\mathcal{G}}(\cdot)} \sum_{t=0}^{\infty} \mathbb{E}_{s_t \sim \rho_{\bar{\pi}}^t(\cdot | g), g' \sim \eta_{\bar{\pi}}^t(\cdot | g)} \gamma^t D_{TV}(\bar{\pi}(\cdot \mid s, g') \| \pi_{\theta}(\cdot \mid s, g')), \tag{3}$$

for all policy $\pi, \bar{\pi} \in \Pi_{\Theta}$, and goal $g \in \mathcal{G}$, where $D_{TV}(p \| q) := \sum_x |p(x) - q(x)|/2$ is the total variation. Although in behavior cloning, KL divergence is usually adopted as the distance of polices, minimizing KL divergence is also equivalent to minimize total variation by Pinker's Inequality. Then the behavior cloning process corresponds to the following optimization w.r.t. $\theta$.

$$\theta^{(k)} = \operatorname*{argmin}_{\theta \in \Theta} F_{\eta}(\pi_{\theta} \| \pi^{(k)}). \tag{4}$$

---

**Algorithm 1** Self-Imitation Policy Learning through Iterative Distillation(SPLID): Meta Algorithm

---

1: Initialize parameter $\theta^{(0)}$ and target policy $\pi^{(0)}$ with uniform distribution.
2: **for** episode $k = 1, ..., K$ **do**
3:     Enhance exploration from $\pi_{\theta^{(k-1)}}$ to generate a new behavior policy $\tilde{\pi}^{(k)}$ with constraint Equation 1.
4:     Sample trajectories from the environment following $\tilde{\pi}^{(k)}(\cdot \mid \cdot, g)$.        *//Exploration*
5:     **if** $\tilde{\pi}^{(k)}$ is a $\delta$-distilled policy (Definition 2) of $\pi_{\theta^{(k-1)}}$ **then**
6:         Mix $\tilde{\pi}^{(k)}$ and $\pi^{(k-1)}$ to obtain $\pi^{(k)}$ via Equation 2.        *//Distill the target*
7:     **else**
8:         Keep target policy unchanged $\pi^{(k)} = \pi^{(k-1)}$.
9:     **end if**
10:    Relabel the target policy $\pi^{(k)}$ and conduct behavior cloning to update $\theta^{(k)}$ via Equation 4.
11:                                                                                    *//Behavior Cloning*
12: **end for**

---

## 3.2 THEORETICAL ANALYSIS

As SPLID is aimed at boosting the sample efficiency, we analyze the theoretical performance of SPLID in this section. To begin with, we assert the following assumption to restrict the relabeling strategy $\eta_\pi^t$, which ensures that relabeling will contribute to the policy learning.

**Assumption 2** (Proper Relabeling). *It holds that*

$$\inf_{g \in supp(p_\mathcal{G}(\cdot))} \mathbb{E}_{g' \sim p_\mathcal{G}(\cdot)} \eta_\pi^t(g \mid g') \geq C_\pi > 0,$$

*for all policy $\pi \in \Pi_\Theta$, $t \in \mathbb{N}$ and an absolute constant $C_\pi$ only relevant with $\pi$, where supp($\cdot$) is the support set.*

Assumption 2 is easy to be satisfied when the relabeling strategy is selected as random sampling with a certain probability, which has been applied in practice. Another usual method is to use the visitation measure as the relabeling strategy, for example, HER applies the uniform future state relabeling strategy, i.e.,$\eta_\pi^t(\cdot \mid g) \propto \sum_{i=t}^\infty \gamma^i \rho_\pi^i(\cdot \mid g)$. When $\pi$ is near optimal, the relabeling strategy $\eta_\pi^t$ of HER also satisfies this assumption.

Based on the assumption, the following theorem shows the local convergence of Algorithm 1.

**Theorem 1** (Local Convergence). *For a multi-goal MDP with infinite horizon, under Assumption 2, it holds for Algorithm 1 that,*

$$|J(\pi^\star) - J(\pi_{\theta^{(k)}})| \leq \frac{2}{(1-\gamma)^2} \Delta_k + |J(\pi^\star) - J(\pi^{(k)})|,$$

*for all policies $\pi^\star \in \Pi_\Theta$ and $k \in [K]$, where $\Delta_k$ is defined as*

$$\Delta_k := C_{\pi_{\theta^{(k)}}}^{-1} F_\eta(\pi_{\theta^{(k)}} \| \pi^{(k)}),$$

*where $C_{\pi_{\theta^{(k)}}}$ and $\epsilon$ are specified in Assumption 2 and Equation 1 respectively.*

*Proof of Theorem 1.* See detailed proof in Appendix B.1. □

Theorem 1 reveals that Algorithm 1 can converge to the local optimality if the locally optimal policy $\pi^\star$ has the same discounted return with $\pi^{(k)}$ and the surrogate function is minimized to zero. Furthermore, we establish the following theorem to show the guarantee of monotonicity on the discounted return.

**Theorem 2** (Policy Improvement Guarantee). *For a multi-goal MDP with infinite horizon, under Assumption 1 and 2, it holds for Algorithm 1 that ,*

$$J(\pi_{\theta^{(k)}}) - J(\pi_{\theta^{(k-1)}}) \geq \delta - \frac{2}{(1-\gamma)^2} [(1-\alpha_k)(\Delta_{k-1} + \epsilon) + \Delta_k], \tag{5}$$

*where $\Delta_k$ is defined in Theorem 1, $\epsilon$ is specified in Equation 1, and $\alpha_k$ is defined in Equation 2.*

*Proof of Theorem 2.* See detailed proof in Appendix B.1. □

We motivate the proof by the following decomposition.

$$J(\pi_{\theta^{(k)}}) - J(\pi_{\theta^{(k-1)}}) = \underbrace{J(\pi_{\theta^{(k)}}) - J(\pi^{(k)})}_{(i)} + \underbrace{J(\pi^{(k)})) - J(\tilde{\pi}^{(k)})}_{(ii)} + \underbrace{J(\tilde{\pi}^{(k)}) - J(\pi_{\theta^{(k-1)}})}_{(iii)}. \tag{6}$$

Term (i) in Equation 6 corresponds to the performance gap from the behavior cloning and is a direct conclusion of Theorem 1. Term (ii) in Equation 6 matches how $\pi^{(k)}$ is mixed by $\pi^{(k-1)}$ and $\tilde{\pi}^{(k)}$, although this term requires more steps to control. Term (iii) in Equation 6 reflects how $\delta$- distilled policy $\tilde{\pi}^{(k)}$ contributes to the performance improvement.

From Theorem 2, it suggests that once the surrogate function can be minimized near to zero and the exploration rate is controlled, then the positive $\delta$ on the RHS of Equation 5 will dominate, ensuring the monotonicity of the return.

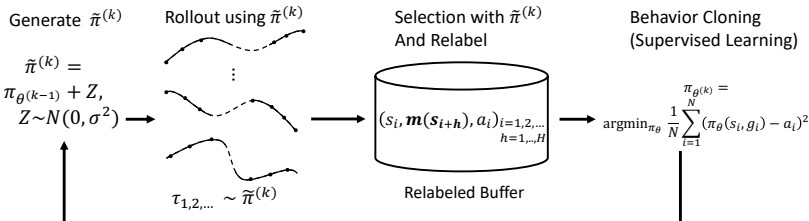

Figure 1: Protocol of Self-Imitation Policy Learning through Iterative Distillation (SPLID) in the practical implementation (Algorithm 2).

## 3.3 PRACTICAL IMPLEMENTATION

In this section, we desire to propose a practical implementation of the meta-algorithm SPLID (Algorithm 1). Although Algorithm 1 has the theoretical performance improvement and local convergence guarantee, the selection of the $\delta$-distilled policy (Line 3 of Algorithm 1) and how to generate a new behavior policy (Line 5 of Algorithm 1) require detailed implementations. To this end, we consider instantiating Algorithm 1 in the case where the transition kernel of the environment $\mathcal{T}$ and all the policies in $\Pi_\Theta$ are deterministic and we show that we are able to select such $\delta$-distilled policy in a practical method by the following proposition, which characterizes the relationship between $\delta$-distilled policy and First Hit Time (FHT).

**Proposition 1.** *Suppose the transition kernel of the environment $\mathcal{T}$ and all the policy in $\Pi_\Theta$ are deterministic and Assumption 1 holds. Then $\tilde{\pi}$ is the $\delta$-distilled policy of $\pi$, if $\tilde{\pi}$ achieves goals with strictly less FHT than $\pi$, where FHT is defined as $FHT(\pi, g) := \min\{t \in \mathbb{N} : m(s_t) = g, s_t \sim \rho_\pi^t\}$.*

*Proof of Proposition 1.* See proof in Appendix B.3. □

Based on Proposition 1, we instantiate Algorithm 1 to a concrete algorithm in Algorithm 2 (see full algorithm in Appendix A), which also achieves empirical success in the continuous environments with mild stochasticity. The whole pipeline of Algorithm 2 is illustrated in Figure 1.. For simplicity, we also name Algorithm 2 SPLID and we introduce three main stages of it for each episode as follows.

**Stage 1: Exploration with a New Policy** SPLID utilizes the idea of parameter noise (Plappert et al., 2017) to enhance exploration to generate new behavior policy. Specifically, SPLID maintains an deterministic policy $\pi_{\theta_{k-1}}$ at the $(k-1)$-th episode, parameterized as a policy network. In the $k$-th episode a new stochastic behavior policy $\tilde{\pi}^{(k)}$ is generated by

$$\tilde{\pi}^{(k)} = \pi_{\theta(k-1)} + Z, Z \sim \mathcal{N}(0, \sigma^2). \tag{7}$$

According to Equation 7, the behavior policy $\tilde{\pi}^{(k)}$ comes from adding a Gaussian exploration noise upon the old model policy $\pi_{\theta(k-1)}$, as in previous deterministic policy learning literature (Lillicrap et al., 2015; Fujimoto et al., 2018). Concretely, during training, $\tilde{\pi}^{(k)}$ first interacts with the environment and collects a batch of transition samples, permitting us to generate a batch of state-goal-action pairs, regardless of their optimality. These state-goal-action pairs contain a set of transition tuples $(s, g', a)$, where $g'$ denotes the hindsight goal. i.e.,the starting point, finally achieved the goal, and the corresponding actions are included in each of these transition tuples. From an oracle-perspective, these state-action pairs can be regarded as ones generated from a series of *unknown deterministic policies* instead of a known stochastic behavior policy $\tilde{\pi}^{(k)}$ and each pair provides an individual solution for the state-goal pair $(s, g')$ task. Such a method enables us to obtain more alternatives for the later selection process, which increases the sample efficiency. Owing to the Equation 7, the constraint in Equation 1 between $\pi_{\theta(k-1)}$ a nd $\tilde{\pi}^{(k)}$ is naturally satisfied with a high probability since $\epsilon$ in Equation 1 is upper bounded with the polynomial of $\sigma$ with a high probability.

**Stage 2: Select and Relabel** As the instantiation of Line 5 in Algorithm 1 into Algorithm 2, by Proposition 1, SELECT function chooses $\delta$-distilled policy by comparing the FHT. To this end, we empirically apply a SELECT function to distinguish FHT and store the fragments of the policy

with strictly less FHT in a buffer $B = \{(s_i, g_i', a_i)\}_{i=1,2,...,N}$,. The SELECT function can be implemented by resetting the environment to a given previous state, which is always tractable in simulation (Nair et al., 2018), Specifically, SELECT function takes in an episode generated by $\tilde{\pi}^{(k)}$. Suppose the episode $(s_t, a_t, s_{t+1}, a_{t+1}, ..., s_{t+h})$ is of length $h$, the SELECT function resets environment to the starting state of this episode $s_t$ and runs $\pi_{\theta^{(k-1)}}$ for up to $k$ steps, trying to reach the final achieved state $s_{t+k}$. i.e.,, at every step, an action of $\pi_{\theta^{(k-1)}}(s, m(s_{t+h}))$ is performed. If $\pi_{\theta^{(k-1)}}$ is NOT able to reach $s_{t+k}$ within $k$ steps, the corresponding transition tuple $(s_t, m(s_{t+h}), a_t)$ will be collected in the buffer $B$ and we will update $\theta$ from these tuples later. As horizons in the practical tasks are usually finite, we truncate the horizon by $T$. While in practice, we apply a notion of learning horizon $H \leq T$, that is, during selection process we also truncate FHT with $H$,i.e.,$FHT = \min\{H, FHT\}$. The introduction of $H$ is a necessary trade-off, which will be explained from the perspectives of both the theory and ablation study in Section 4.1.

**Stage 3: Behavior Cloning** Then, we can apply behavior cloning to update $\pi_\theta$ following the same procedure as in Algorithm 1. To be specific, we use supervised learning to minimize the Mean Squared Error between the action stored in the buffer and the action $\pi_\theta$ predicted as follows

$$\pi_\theta = \arg\min_{\pi_\theta \in \Pi_\Theta} \frac{1}{N} \sum_{i=1}^{N} (\pi_\theta(s_i, g_i') - a_i)^2, \tag{8}$$

where $N$ is the mini-batch size , $(s_i, g_i', a_i)$ is sampled from the buffer $B$ and $\pi_\theta(s, g) := \{a \in \mathcal{A} : \pi_\theta(a \mid s, g) = 1\}$. From this point of view, the SPLID method is also composed of evolution strategy and policy distillation, where a stochastic target policy $\tilde{\pi}^{(k)}$ acts as the perturbation on the action space and produces diverse strategies (*a population*), and we choose those well-performed strategies to distill their knowledge into $\pi_\theta$ (a *selection*).

Moreover, although plenty of policy gradient methods leverage action space noise in exploration Lillicrap et al. (2015); Schulman et al. (2017); Fujimoto et al. (2018), there is a clear gap between those perturbations and the evolutionary insight in our methods: The most important difference between action space noise in SPLID and previous policy gradient methods is that SPLID regards those *stochastic variants* as from a series of unknown *oracle deterministic policies*, then use a selection function as a filter the well-performed unknown oracle policies and finally distill those transitions to the model policy. On the other hand, previous policy gradient methods do not contain such a selection step, but instead, use *all* those variants to estimate the value function and improve the policy through policy gradient accordingly.

# 4 EXPERIMENTS

## 4.1 RESULT ON THE GOAL CONDITIONED TASKS BENCHMARKS

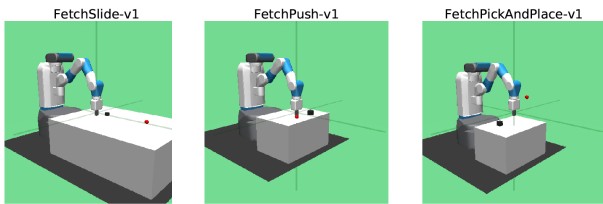

Figure 2: Three goal-oriented tasks involved in this paper.

We demonstrate the proposed method on the Goal-Conditioned task, Fetch benchmarks (Plappert et al., 2018). Specifically, we evaluate our method on the FetchPush, FetchSlide, and FetchPickAndPlace (see Figure 2 for illustration and Plappert et al. (2018) for detailed description). These three environments are nearly deterministic, satisfying the condition of Proposition B.3. They are also resettable so that we can apply SELECT function in Algorithm 2. We compare our proposed method with the HER (Andrychowicz et al., 2017; Plappert et al., 2018) released in OpenAI Baselines (Dhariwal et al., 2017). We also include PCHID (Sun et al., 2019), GCSL (Ghosh et al., 2019b), and Evolution-

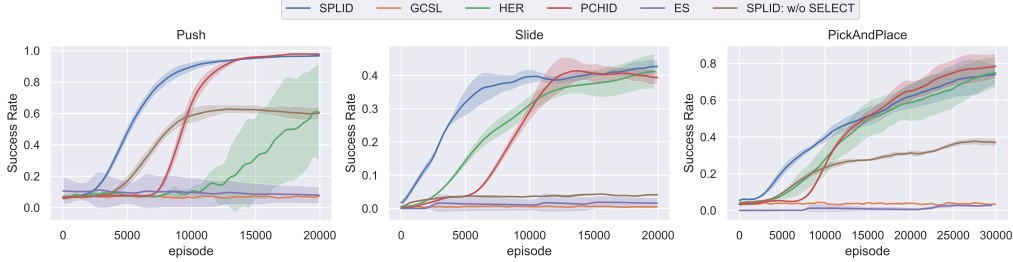

Figure 3: Test performance (Success Rate) of SPLID (our proposed method), GCSL, PCHID, HER, ES, and SPLID without SELECT function. Results are averaged under 5 random seeds.

ary Strategy (ES) (Salimans et al., 2017) as the baseline. To show the effect of SELECT function, we include the ablation study of SPLID without SELECT function.

Figure 3 shows the comparison of different approaches, where we adopt the success rate as the metric of performance. For each environment, we conduct 5 experiments with different random seeds and plot the averaged learning curve. SPLID shows superior learning efficiency and can learn to solve the task in fewer episodes in all the three environments, compared with HER, GCSL, PCHID, and ES. Ablation study on the SELECT function of SPLID reveals the indispensable role of SELECT function in distilling the goal and reaching high sample efficiency and corresponds to the theoretical analysis in Section 3.2. In Appendix D, we also plot the performance of the ablation study on SELECT function in metric of the average return, which demonstrates how SELECT function plays a role in searching for $\delta$-distilled policy and improves the quality of the target of behavior cloning.

We also observe that GCSL fails in all these three environments, even performing worse than SPLID without SELECT function. Because of the difficulty of these three benchmarks, only relabeling to the last step like GCSL is inadequate for effective learning. On the contrary, SPLID relabels with all future states with selective relabeling strategy and leverages the qualify of the target policy, making it able to cope with the complicated tasks. We conduct more experiments where GCSL has non-trivial performance and SPLID still outperforms GCSL on these environments. The details and results are in Appendix E.

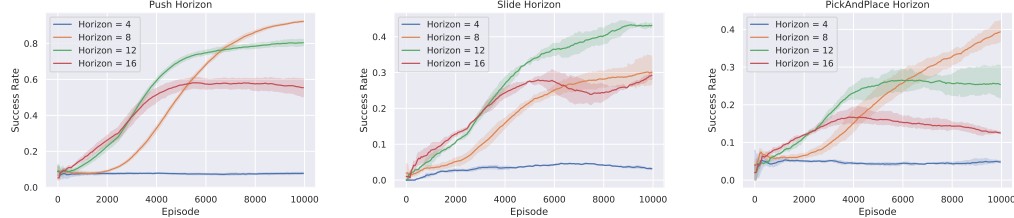

Figure 4: Ablation study on learning horizon H.

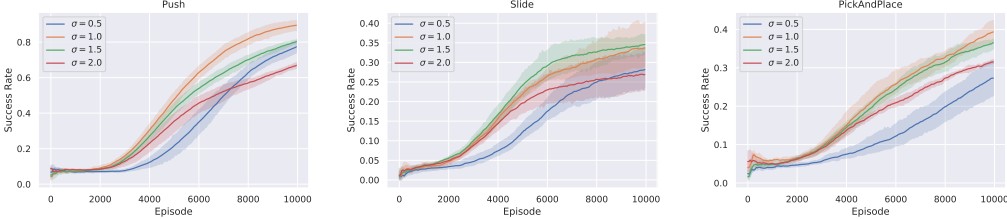

Figure 5: Ablation study on exploration factor $\sigma$.

**Exploration Factor** The exploration factor $\sigma$ controls the randomness of behavior policy $\tilde{\pi}^{(k)}$ and therefore determines the behavior of generated samples. While larger $\sigma$ helps the agents to benefit

exploration by generating samples with large variance, smaller $\sigma$ helps to generate a biased sample with little variance. Here we need to select a proper $\sigma$ to balance the variance and bias and help achieve higher performance guarantee in Theorem 2, where $\epsilon$ is determined by exploration factor $\sigma$. Figure 5 shows our ablation study on the selection of different exploration factors. The results are generated with 5 different random seeds. We find in all environments, the exploration factor $\sigma = 1$ provides sufficient exploration and relatively high learning efficiency.

**Learning Horizon**   In our proposed method, the parameter of learning horizon $H \leq T$ determines the maximal length of sample trajectories the policy can learn from. Intuitively, smaller $H$ decreases the learning efficiency as the policy is limited by its small horizon, making it hard to plan for the tasks that need more steps to solve. Strictly, when $H$ is smaller, then the $\delta$ in Theorem 2 is higher and the stricter selection makes the buffer contain less data. On the other hand, larger $H$ will provide a better concept of the local as well as global geometry of the state space, and thus the agent may learn to solve more challenging tasks. Also, large $H$ suggests lower $\delta$ but probably higher $\alpha_k$ in Theorem 2, meaning we have a larger buffer but with perhaps lower quality as the target of behavior cloning. Another issue of using larger $H$ is the introduction of more interactions with the environment to gain enough buffer data with the need of more computation time. Moreover, as the tasks normally do not need lots of steps to finish, when the learning horizon is getting too large, more noisy actions will be collected and be considered as better solutions and hence impede the learning performance. Figure 4 shows our ablation studies on the selection of learning horizon $H$. The results are generated with 5 different random seeds. We find that $H = 8$ provides satisfactory results in all of the three environments.

It is worth noting that PCHID can be seen as a special case of SPLID if we gradually increase the learning horizon in SPLID from 1 to $H$. From this perspective, as shown in Figure 4 where at the beginning of learning a smaller Horizon leads to poor performance, the performance of PCHID is upper bounded by SPLID.

## 5   RELATED WORK

**Supervised and Self-Imitate Approaches in RL.**   Recently, several works put forward to use supervised learning to improve the stability and efficiency of RL. Zhang et al. (2019) propose to utilize supervised learning to tackle the overly large gradients problem in policy gradient methods. In order to improve sample efficiency, the work chose to first design a target distribution proposal and then used supervised learning to minimize the distance between present policy and the target policy distribution. The Upside-Down RL proposed by Schmidhuber (2019) used supervised learning to mapping states and rewards into action distributions, and therefore acted as a normal policy in RL. Their experiments show that the proposed UDRL method outperforms several baseline methods Srivastava et al. (2019). In the work of Sun et al. (2019), a curriculum learning scheme is utilized to learn policies recursively. The self-imitation idea relevant to SPLID is also discussed in the concurrent work of Ghosh et al. (2019a), but SPLID further uses the SELECT function to improve the quality of collected data for self-imitation learning. Ghosh et al. (2019b) propose a new algorithm GCSL applying behavior cloning methods in multi-goal tasks with relabelling methods and SPLID is aimed at improving the sample efficiency of their algorithm through leveling up the quality of the target policy to imitate on. Due to the page limit, we postpone the remaining discussions on related works in Appendix C.

## 6   CONCLUSION

This work proposes a novel meta-algorithm Self-Imitation Policy Learning through Iterative Distillation (SPLID), which relies on the concept of $\delta$-distilled policy to iteratively level up the quality of the target and relabelled data that the agent mimics. From the perspective of theory, we study the local convergence and monotonic performance improvement guarantee, under certain assumptions. We also implement a practical version of SPLID and test it on the Multi-Goal Benchmarks. Experiments show that SPLID outperforms other goal goal-oriented RL methods with a substantial margin, which also corresponds to the theoretical analysis. Since meta-algorithm SPLID shares good theoretical properties, it is worthwhile in the future work to study how to instantiate it in other practical manners.

## REPRODUCIBILITY STATEMENT

For the theoretical work in this paper, all the assumptions have been clarified immediately after the occurrence and all the detailed proofs of theorems can be found in the Appendix. For the empirical work, we submit a zip of Jupyter Notebooks as supplementary material, where we record all the codes used for the experiments. All these efforts guarantee the reproducibility of our work.

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

## A    PESUDOCODE: PRACTICAL IMPLEMENTATION OF SPLID

---

**Algorithm 2** SPLID: Practical Implementation

---

1: **Require**
2:   1.A reward function $r(s, g) = 1$ if $g = m(s)$ else 0.
3:   2. Learning Horizon $H$.
4: Initialize replay buffer $B = \{\}$ and $\theta^{(0)}$.
5: **for** episode $k = 1, ..., K$ **do**
6:     Generate a new policy $\tilde{\pi}^{(k)} = \pi_{\theta^{(k-1)}} + Z, Z \sim \mathcal{N}(0, \sigma^2)$                    *//Exploration*
7:     Generate initial state $s_0$ and goal $g$ by the environment
8:     **for** $t = 1, ..., T$ **do**
9:         Select an action by the behavior policy $a_t = \tilde{\pi}^{(k)}(s_t, g)$
10:        Execute the action $a_t$ and get the next state $s_{t+1}$
11:    **end for**
12:    **for** $t = 1, ..., T$ **do**
13:        **for** $h = 1, ..., H$ **do**
14:            Calculate additional goal according to $s_{t+h}$ by $g' = m(s_{t+h})$
15:            **if** SELECT$(s_t, g')$ = True **then**
16:                Store $(s_t, g', a_t)$ in $B$                    *//Distill the target*
17:            **end if**
18:        **end for**
19:    **end for**
20:    Sample a minibatch from buffer $B$
21:    Do supervised learning to update $\theta$ via Equation 4.                    *//Behavior Cloning*
22: **end for**

---

## B    DETAILED PROOF

### B.1    PROOF OF THEOREM 1

*Proof.* To prove Theorem 1, we first introduce one lemma to bridge the gap between the behavior cloning and performance difference.

**Lemma 1.** *For a multi-goal MDP with infinite horizon, it holds that*

$$|V(\pi, g) - V(\pi^\star, g)| \leq \frac{2\|r(\cdot, g)\|_\infty}{(1-\gamma)^2} \mathbb{E}_{s \sim \rho_\pi(\cdot|g)}[D_{TV}(\pi^\star(\cdot \mid s, g)\|\pi(\cdot \mid s, g))],$$

*where for every goal $g \in \mathcal{G}$ and policies $\pi, \pi^\star \in \Pi_\Theta$. Here $D_{TV}(p\|q)$ is the total variation between two probability measures $p$ and $q$, and $\rho_\pi(\cdot \mid g)$ is the discounted average state visitation measure induced by policy $\pi$, and $\|\cdot\|_\infty$ is the $\infty$-norm.*

*Proof.* It is a direct corollary of one-goal MDP case in Theorem 5.1 of Xu et al. (2019).    □

To ease the reading, we define $\bar{\pi} = \pi_{\theta^{(k)}}$ in the proof. By the definition of surrogate function in Equation 3, it holds that

$$F_\eta(\pi_\theta\|\bar{\pi}) = \mathbb{E}_{g \sim p_\mathcal{G}(\cdot)} \sum_{t=0}^\infty \mathbb{E}_{s_t \sim \rho_{\bar{\pi}}^t(\cdot|g), g' \sim \eta_{\bar{\pi}}^t(\cdot|g)} \gamma^t D_{TV}(\bar{\pi}(\cdot \mid s, g')\|\pi_\theta(\cdot \mid s, g'))$$

$$\geq \mathbb{E}_{g \sim p_\mathcal{G}(\cdot)} \sum_{t=0}^\infty \mathbb{E}_{s_t \sim \rho_{\bar{\pi}}^t(\cdot|g), g' \sim \eta_{\bar{\pi}}^t(\cdot|g)} \gamma^t \mathbb{I}(g = g') D_{TV}(\bar{\pi}(\cdot \mid s, g')\|\pi_\theta(\cdot \mid s, g'))$$

$$\geq C_{\bar{\pi}} \mathbb{E}_{g \sim p_\mathcal{G}(\cdot)} \sum_{t=0}^\infty \mathbb{E}_{s_t \sim \rho_{\bar{\pi}}^t(\cdot|g)} \gamma^t D_{TV}(\bar{\pi}(\cdot \mid s, g)\|\pi_\theta(\cdot \mid s, g))$$

$$\geq C_{\bar{\pi}} \mathbb{E}_{g \sim p_\mathcal{G}(\cdot)} \mathbb{E}_{s \sim \rho_{\bar{\pi}}(\cdot|g)} D_{TV}(\bar{\pi}(\cdot \mid s, g)\|\pi_\theta(\cdot \mid s, g)).$$

Applying Lemma 1, it yields that

$$
\frac{2}{(1-\gamma)^2} F_\eta(\pi_\theta \| \bar\pi) \geq C_{\bar\pi} \mathbb{E}_{g \sim p_\mathcal{G}(\cdot)} |V(\pi_\theta, g) - V(\bar\pi \mid g)|)
$$
$$
\geq C_{\bar\pi} |\mathbb{E}_{g \sim p_\mathcal{G}} (V(\pi_\theta, g) - V(\bar\pi \mid g))|
$$
$$
= C_{\bar\pi} |J(\pi_\theta) - J(\bar\pi)|,
$$

where the first inequality follows from the fact that $|r(s, g)| \leq 1$ for all $(s, g) \in \mathcal{S} \times \mathcal{G}$ and the last inequality relies on the fact that absolute function $|\cdot|$ is convex. Then we prove Theorem 1 by taking additional triangle inequality. $\qquad\square$

### B.2 Proof of Theorem 2

*Proof.* Recall the motivated decomposition in Equation 6 as follows,

$$
J(\pi_{\theta^{(k)}}) - J(\pi_{\theta^{(k-1)}}) = \underbrace{J(\pi_{\theta^{(k)}}) - J(\pi^{(k)})}_{\text{(i)}} + \underbrace{J(\pi^{(k)})) - J(\tilde\pi^{(k)})}_{\text{(ii)}} + \underbrace{J(\tilde\pi^{(k)}) - J(\pi_{\theta^{(k-1)}})}_{\text{(iii)}}. \quad (9)
$$

Applying Theorem 1 with $\pi^\star = \pi^{(k)}$, then we immediately obtain the desired result of term (i) in Equation 9. Then it suffices to bound the next two terms.

**Term (ii) in Equation 9**    By the mixture rule in Equation 2 , we have that

$$
\begin{aligned}
J(\pi^{(k)}) - J(\tilde\pi^{(k)}) &= ((1 - \alpha_k) J(\pi^{(k-1)}) + \alpha_k J(\tilde\pi^{(k)})) - J(\tilde\pi^{(k)}) \\
&= (1 - \alpha_k)(J(\pi^{(k-1)}) - J(\tilde\pi^{(k)})) \\
&= (1 - \alpha_k)\Big( \underbrace{J(\pi^{(k-1)}) - J(\pi_{\theta^{(k-1)}})}_{\text{(iia)}} + \underbrace{J(\pi_{\theta^{(k-1)}}) - J(\tilde\pi^{(k)})}_{\text{(iib)}} \Big).
\end{aligned} \quad (10)
$$

First we deal with term (iia) in Equation 10, and we observe that term (iia) has the similar forms with term (i) in Equation 9 but differs with index $k$. Hence we bound term (iia) by applying Theorem 1. It suggests that

$$
\text{(iia)} \geq -C_{\pi^{(k)}} \frac{2}{(1-\gamma)^2} F_\eta(\pi_{\theta^{(k)}} \| \pi^{(k)}).
$$

As for the term (iib) in Equation 10, recall that the $\tilde\pi^{(k)}$ is generated by $\pi_{\theta^{(k-1)}}$ with constraint in Equation 1. Again we applying Lemma 1, we derive that

$$
\begin{aligned}
\text{(iib)} &\geq |J(\pi_{\theta^{(k-1)}}) - J(\tilde\pi^{(k)})| \\
&= -\left| \mathbb{E}_{g \sim p_\mathcal{G}(\cdot)} \big( V(\pi_{\theta^{(k-1)}}, g) - V(\tilde\pi^{(k)}, g) \big) \right| \\
&\geq -\mathbb{E}_{g \sim p_\mathcal{G}(\cdot)} \left| \big( V(\pi_{\theta^{(k-1)}}, g) - V(\tilde\pi^{(k)}, g) \big) \right| \\
&\geq -\frac{2}{(1-\gamma)^2} \sup_{s,g} D_{TV}(\pi_{\theta^{(k-1)}}(\cdot \mid s, g) \| \tilde\pi^{(k)}(\cdot \mid s, g)) \\
&\geq -\frac{2\epsilon}{(1-\gamma)^2},
\end{aligned}
$$

where the last inequality holds due to Equation 1. Hence, we lower bound term (ii) in Equation 10 as follows.

$$
\text{(ii)} \geq -\frac{2(1 - \alpha_k)}{(1-\gamma)^2} [C_{\pi^{(k-1)}} F_\eta(\pi_{\theta^{(k-1)}} \| \pi^{(k-1)}) + \epsilon].
$$

**Term (iii) in Equation 9**   To ease the reading, we denote that $\pi := \tilde{\pi}^{(k)}$ and $\bar{\pi} := \pi_{\theta^{(k-1)}}$ in this proof. By the idea of importance sampling and Definition 2, we have that

$$
\begin{aligned}
J(\pi) - J(\bar{\pi}) &= \mathbb{E}_{g \sim p_{\mathcal{G}}(\cdot)}[V(\pi, g) - V(\bar{\pi}, g)] \\
&= \mathbb{E}_{g' \sim p_{\mathcal{G}}(\cdot), g \sim \rho_\pi(m^{-1}(\cdot)|g')} \frac{p_{\mathcal{G}}(g)}{p_{\mathcal{G}}(g')\rho_\pi(m^{-1}(g) \mid g')}[V(\tilde{\pi}, g) - V(\bar{\pi}, g)] \\
&\geq \mathbb{E}_{g' \sim p_{\mathcal{G}}(\cdot), g \sim \rho_\pi(m^{-1}(\cdot)|g')} \frac{p_{\mathcal{G}}(g)}{p_{\mathcal{G}}(g')}[V(\pi, g) - V(\bar{\pi}, g)] \\
&= \sum_{g' \in \mathcal{G}} p_{\mathcal{G}}(g) \mathbb{E}_{g \sim \rho_\pi(m^{-1}(\cdot)|g')}[V(\pi, g) - V(\bar{\pi}, g)] \\
&= G(\pi \| \bar{\pi}) \geq \delta.
\end{aligned}
$$

To sum up, plugging term (i), term (ii), and term (iii) into Equation 9, we finally obtain that

$$
\begin{aligned}
J(\pi_{\theta^{(k)}}) - J(\pi_{\theta^{(k-1)}}) \geq &\ \delta - \frac{2}{(1-\gamma)^2} C_{\pi^{(k)}} F_\eta(\pi_{\theta^{(k)}} \| \pi^{(k)}) \\
&- \frac{2(1-\alpha_k)}{(1-\gamma)^2}[C_{\pi^{(k-1)}} F_\eta(\pi_{\theta^{(k-1)}} \| \pi^{(k-1)}) + \epsilon],
\end{aligned}
$$

which concludes the proof of Theorem 2. □

### B.3   PROOF OF PROPOSITION 1

*Proof.*   Under the conditions of Proposition 1, it holds that

$$
V(\pi, g) = \sum_{i=FHT(\pi,g)}^{\infty} \gamma^i = \gamma^{FHT(\pi,g)}/(1-\gamma)
$$

and $\gamma \in (0, 1)$, then if $FHT(\tilde{\pi}, g) > FHT(\pi, g)$, it holds that $V(\tilde{\pi}, g) > V(\pi, g)$. If $\tilde{\pi}$ achieve goals with strictly less FHT than $\pi$, we have that for all $g \in \mathcal{G}$, $\mathbb{E}_{g \sim \rho_{\bar{\pi}}(m^{-1}(\cdot)|g')}\big[(V(\tilde{\pi}, g) - V(\pi, g))\big] \geq \delta_0 > 0$. By Assumption 1, Proposition 1 holds for certain absolute constant $\delta$.

□

## C   MORE RELATED WORKS

**Evolution Strategies and Parameter Noise.**   The Evolution Strategy (ES) was proposed by Salimans et al. (2017) as an alternative to standard RL approaches, where the prevailing temporal difference based value function updates or policy gradient methods are replaced as perturbations on the parameter space to resemble the evolution. Later on, Campos et al. (2018) improved the efficiency of ES by means of importance sampling. Besides, the method was also extended to be combined with Novelty-Seeking to further improve the performance (Conti et al., 2018). Thereafter, Plappert et al. (2017) proposed to use Parameter Noise as an alternative to the action space noise injection for better exploration. They show such a perturbation on the parameter space can be not only used for ES methods, but also collected to improve the sample efficiency by combining it with traditional RL methods. While previous ES algorithms apply perturbations on the parameter noise and keep the best-performed variates, our approach implicitly execute the policy evolution by distilling better behaviors, therefore our approach can be regarded as an evolutionary method based on action space perturbation.

**Learning with Experts and Policy Distillation.**   Policy Distillation (PD) was proposed to extract the policy of a trained RL agent with a smaller network to improve the efficiency as well as the final performance or combine several task-specific agents together (Rusu et al., 2015). Latter extensions proposed to improve the learning efficiency (Schmitt et al., 2018), enhance multi-task learning (Teh et al., 2017; Arora et al., 2018). All of those methods start from a trained expert agent or human expert experience that can solve a specific task (Czarnecki et al., 2019). As a comparison, our proposed method focus on extracting knowledge from stochastic behaviors, which is capable to act as a feasible policy itself with regard to the primal task. In PD, a deterministic teacher policy is

provided and can be queried to generate enough data to teach the student policy. However, in SPLID all the teacher policies are unknown as they are *stochastic variants* of the student policy. We regard those stochastic variants as they are from *unkonwn deterministic oracle policies* and distill those policies through the single corresponding trajectory. In SPLID, the teacher model is unknown and can not be queried as in PD.

# D   DISCUSSIONS ABOUT $\delta$-DISTILLED POLICY AND SELECT FUNCTION

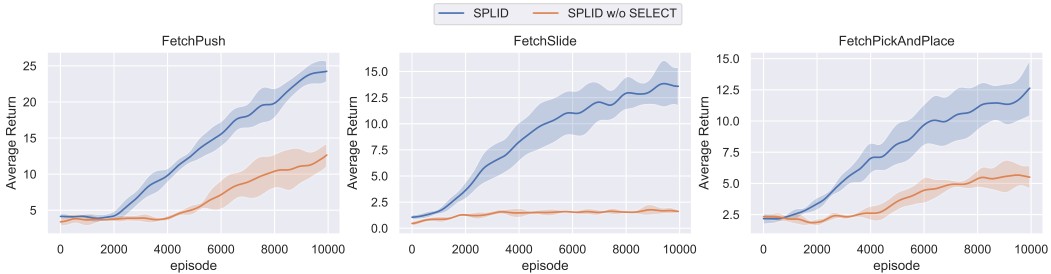

Figure 6: Average Return of SPLID and SPLID w/o SELECT function on Fetch environments. Results are averaged under 5 seeds.

In this section, we discuss the role of $\delta$-distilled policy in the practical implementation of SPLID. Theorem 2 reveals the theoretical guarantees of the performance of SPLID, which relies on the $\delta$-distilled policy. Only a proper and absolute $\delta$ can SPLID enjoy performance improvement guarantee by Theorem 2. In the practical implementation of SPLID, because of Proposition B.3, we pursue such $\delta$-distilled policy by comparing First Hit Time (FHT). With the help of SELECT function , we can distinguish whether $\tilde{\pi}^{(k)}$ has shorter FHT than $\pi_{\theta^{(k-1)}}$ and hence implement Line 5 of meta-algorithm SPLID (Algorithm 1) in a practical manner. To show how the $\delta$-distilled policy selected by SELECT function contributes to our policy learning in practice, we conduct three experiments on Fetch environments and apply average return to show the performance of SPLID and SPLID w/o SELECT function. Here return is calculated by summing over all the rewards agent receives during one episode and we take 50 episodes to average the results. Once the agent reaches in the desired goal in fewer time steps, it will receive higher return. (At each step, the agent will receive reward 1 once it reaches the goal and will receive reward 0 otherwise.) According to figure 6, SPLID still outperforms SPLID w/o SELECT with a substantial margin, revealing the function of $\delta$-distilled policy in improving the qualify (FHT) of trained policy.

# E   MORE DISCUSSIONS ABOUT SPLID AND GCSL

As discussed in Section 4.1, GCSL has trivial performance out of serval reasons. To validate our insight, we conduct GCSL and our proposed method on three new environments: Point2D Four Rooms, Sawyer Door, and Reacher. First we introduce these three environments, which are illustrated in figure 7.

**Point2D-FourRoom-v1**   Point2D-FourRoom-v1 is taken from the open-source environment $multi$ world and it requires the blue point to reach the green circle. The state space $[-5, 5] \times [-5, 5]$ has two dimensions representing Cartesian coordinates of the blue point, and the action space $[-1, 1] \times [-1, 1]$ also has two dimensions meaning the horizontal and vertical displacement. There are four rooms separated by gray walls. The goal space is the same as state space, which means $m(s) = s$. The bule point and the green circle are randomly initialized in the state space. The allowable error $\epsilon$ of reaching goal is the radius of the target circle and is set as 1 . The reward function is defined as:

$$r\left(s_{XY}, g_{XY}\right) = \mathbf{1}\left(\|s_{XY} - g_{XY}\|_2^2 \le \epsilon\right).$$

The maximum frames of an episode is 50.

**SawyerDoor-v0** The SawyerDoor-v0 environment is revised from multi-world. The Sawyer robot is required to open the door to a desired angle. The state space (4-dimensional) consists of the Cartesian coordinates of the Sawyer end-effector and the door's angle. The action space is 3-dimensional and controls the position of the end-effector. The desired goals are uniform from 0 to 0.83 radians. And the reward function is defined as:

$$r\left(s_{XY}, g_{XY}\right) = \mathbf{1}\left(\left|s_{\text{angle}} - g_{\text{angle}}\right| \leq \epsilon\right)$$

where the allowable error $\epsilon$ is set as $0.06$. The maximum frames of an episode is $50$.

**Reacher-v2** The Reacher environment is revised from OpenAI Gym (Dhariwal et al., 2017). States are 11-dimensional, which indicates the angles, the positions, and the velocity of the joints. Actions are 2-dimensional and control the movement of two joints. The goals are 2-dimension representing the expected $XY$ position. And the state-to-goal mapping is $m(s) = s[-3 : -1]$, where the last three dimensions are the XYZ position of the end-effect. The reward function is defined as:

$$r\left(s_{XY}, g_{XY}\right) = \mathbf{1}\left(\left\|s_{XY} - g_{XY}\right\|_2^2 \leq \epsilon\right)$$

where the allowable error $\epsilon$ is set as $0.05$. The maximum frames of an episode is $50$.

On these three new environments, we conduct our proposed method and GCSL respectively and derive the results in figure 8 and figure 9. Although GCSL has non-trivial performance, it is clear that our proposed method SPLID outperforms GCSL in terms of both success rate and average return.

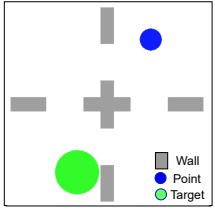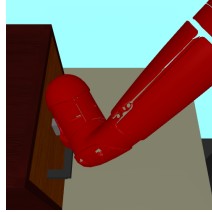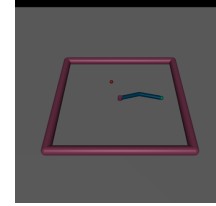

Figure 7: Three new environments: Point2D Four Rooms, Sawyer Door, and Reacher.

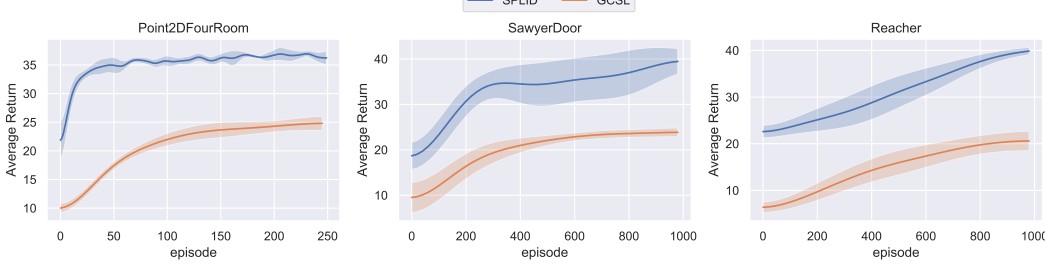

Figure 8: Average Return of SPLID and GCSL on three new environments. Results are averaged under 5 seeds.

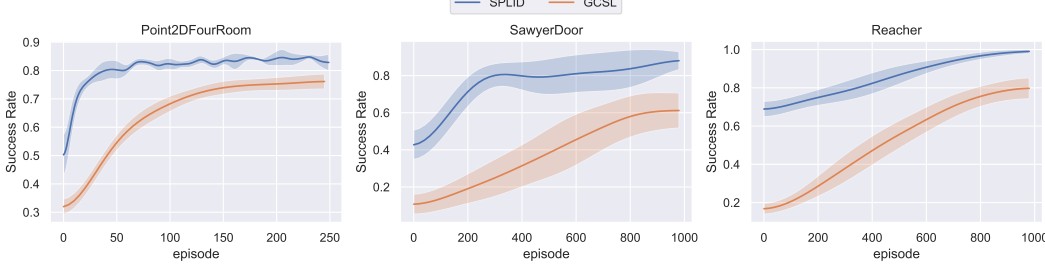

Figure 9: Success Rate of SPLID and GCSL on three new environments. Results are averaged under 5 seeds.

