# OpenReview forum: "SPLID: Self-Imitation Policy Learning through Iterative Distillation"
_ICLR.cc/2022/Conference — ICLR 2022 Submitted_

### Official Review · Reviewer_xVN8 · 2021-11-02

**Correctness:** 3
**Technical Novelty And Significance:** 2
**Empirical Novelty And Significance:** 2
**Recommendation:** 3
**Confidence:** 4

**Main Review:**

Strengths:

- I appreciate the theoretical analysis of the method, as well as the step-by-step walk-through of the algorithm in section 3.1.

Weaknesses:

- Despite similarity with GCSL, GCSL performs far worse than SPLID or even SPLID without SELECT, which as far as I'm concerned is pretty much the same algorithm as GCSL with different exploration (GCSL mentions they initialize their replay buffer with random actions and they play around with different exploration strategies in section A.7, so maybe SPLID w/o SELECT could be considered the same as GCSL). Is the main difference between these methods exploration?
- The paper makes a bold claim that "resetting the environment to a given previous state ... is always tractable in simulation". This seems like a major assumption and is more than the baseline methods assume. Is there a way to implement SPLID without the ability to reset the environment to arbitrary states, perhaps using value functions?
- GCSL only optimizes for reaching the goal state by its final step, which is a slightly different problem setting than this paper. The theoretical improvement that SPLID offers is that, through the SELECT function, it will be able to learn to achieve goals in a shorter number of time-steps. However, the experiments only measure success rate, which does not take into account how optimal these polices are in terms of time taken to reach the goal.
- The paper claims that SPLID does parameter space exploration, but they simply add noise to the actions of the previous policy, which is not parameter space exploration.
- There is no discussion on the assumptions (deterministic, resettable environment) that SPLID makes in relation to the baselines.

My main questions are as follows:

- Why does GCSL perform so poorly?
- Can you implement SPLID without the assumptions of the environment?
- Does SPLID w/ select learn policies that reach the goal in fewer time-steps than SPLID w/o select?

Suggestions (not taken into account for the score):

There are several grammar mistakes as well as other mistakes which caused some confusion (for example, in theorem 2, readers are pointed to the wrong spot to find the definition of $\Delta_k$).

**Summary Of The Paper:**

This paper expands on prior works in self-imitative goal-conditioned RL by proposing to selectively filter the target policy which is used during the behavioral cloning phase. By ensuring that the target policy reaches goals more optimally than the behavior policy, the paper shows that under certain conditions, performance improvement can be guaranteed. The algorithm first collects trajectories under some exploration policy, then for fragments of these trajectories, the algorithm evaluates the current policy (starting in the first state in the fragment) to see if the fragment represents an improvement on the agent's ability to reach the state. If so, it is added to the replay buffer for behavioral cloning. In experiments, the algorithm (SPLID) performs well in deterministic, goal-conditioned environments.

**Summary Of The Review:**

While this paper seems to get strong performance in the selected experiments, I find the experiments to be a bit suspicious and under-explored and I am unconvinced that the performance gained with this method are worth the strong assumptions (env is resettable to a selected state). I cannot recommend accepting this paper.

---

> ### Author Response · Authors · 2021-11-19
> **Response to Reviewer xVN8**
>
> Thank you for your comment. We make the following clarifications to address your concerns on the weakness of this paper respectively:
>
> **Q1:**  *Despite similarity with GCSL, GCSL performs far worse than SPLID or even SPLID without SELECT, which, as far as I'm concerned, is pretty much the same algorithm as GCSL with different exploration (GCSL mentions they initialize their replay buffer with random actions and they play around with different exploration strategies in section A.7, so maybe SPLID w/o SELECT could be considered the same as GCSL). Is the main difference between these methods of exploration?*
>
> **A1:** As we have stated in Section 4.1 in the original paper: Because of the difficulty of these three benchmarks, only relabeling to the last step like GCSL is inadequate for effective learning. On the contrary, SPLID relabels with all future states with selective relabeling strategy and leverages the qualify of the target policy, making it able to cope with the complicated tasks.
> We also add three experiments in Appendix E, the revised paper. In these three environments, GCSL demonstrates non-trivial performance, but our proposed method still achieves better performance in these three environments.
>
> **Q2:** *The paper makes a bold claim that "resetting the environment to a given previous state ... is always tractable in simulation". This seems like a major assumption and is more than the baseline methods assume. Is there a way to implement SPLID without the ability to reset the environment to arbitrary states, perhaps using value functions?*
>
> **A2:** This assumption of a resettable environment is not a *major* assumption of the whole paper, since we need not rely on this assumption to prove any theorems and lemmas in Section 3.2 in the original paper. This assumption only helps us to design practical implementations. Searching for more practical implementations based on this meta-algorithm is our future direction, as stated in the discussion part in the original paper.
>
> **Q3:** *GCSL only optimizes for reaching the goal state by its final step, which is a slightly different problem setting than this paper. The theoretical improvement that SPLID offers is that, through the SELECT function, it will be able to learn to achieve goals in a shorter number of time steps. However, the experiments only measure success rate, which does not take into account how optimal these policies are in terms of time steps taken to reach the goal.*
>
> **A3:** We add experiments in Appendix D and Appendix E in the revised paper for your reference, which shows our proposed method SPLID can learn to achieve goals in a shorter number of time-steps with the help of $\delta$-distilled policy.
>
> **Q4:**  *The paper claims that SPLID does parameter space exploration, but they simply add noise to the actions of the previous policy, which is not parameter space exploration.*
>
> **A4:**  We are only motivated by the ideas of parameter space exploration to specify our protocol to enhance exploration. It is also OK to use other manners to enhance exploration once the constraint is satisfied.
>
> **Q5:** *There is no discussion on the assumptions (deterministic, resettable environment) that SPLID makes in relation to the baselines.*
>
> **A5:** Thank you for your reminder. Baselines (Fetch Environments) are all nearly deterministic and resettable. We add this discussion in the revised paper.
>
> As for your main questions, we answer as follows:
>
> **Q6:** *Why does GCSL perform so poorly?*
>
> **A6:** This question is similar to your first concern, which is answered in A1.
>
> **Q7:** *Can you implement SPLID without the assumptions of the environment?*
>
> **A7:** As we stated in the discussion part in the original paper, our main contribution of this paper is to propose a meta-algorithm and find one practical implementation. Searching for more practical implementations based on this meta-algorithm is our future direction.
>
> **Q8:** *Does SPLID w/ select learn policies that reach the goal in fewer time-steps than SPLID w/o select?*
>
> **A8:** SPLID w/ select Indeed learns policies that reach the goal in fewer time-steps than SPLID w/o select, demonstrated in the experiments in Appendix D in the revised paper.

---

> > ### Comment · Reviewer_xVN8 · 2021-11-29
> > **Response**
> >
> > Thank you for your response. I understand a bit better now the differences between SPLID and GCSL. However, I still am not satisfied with the experiment section. Particularly, I believe Q5 was misunderstood. The algorithms that you compare seem to make different assumptions about the environments that they will be run on, where SPLID exploits the ability to reset the environment arbitrarily during training where the others do not. Intuitively, the ability to reset the environment arbitrarily during training may be a large advantage.
> >
> > I cannot recommend acceptance due to my concerns about the experiment section and also some general concern about the clarity of the paper.

---

### Official Review · Reviewer_FXhj · 2021-11-02

**Correctness:** 2
**Technical Novelty And Significance:** 3
**Empirical Novelty And Significance:** 2
**Recommendation:** 3
**Confidence:** 4

**Main Review:**

Strength:
1. The method is simple yet effective.
2. Theoretical analysis of local convergence and policy improvement is provided.
3. The paper is well written.

Weaknesses:
1. The SELECT requires resetting the environment to a given previous state, which can only be achieved in simulator but not real-world.
2. The overall learning process is similar to Go-Explore. But Go-Explore is not compared.
3. SPLID w/o SELECT does not perform well, which suggests that the method highly relies on the ability to reset the simulator.
4. It is unclear to me why it is called "meta-algorithm".
5. The experiments are weak. The role of $\delta$-distilled policy is unclear. Perhaps some case studies on the identified $\delta$-distilled policy could facilitate the understanding of the algorithm.
6. It is unclear how the algorithm improves the quality of "the timesteps taken to achieve the goal".


**Summary Of The Paper:**

This paper presents a self-imitation learning method for goal-conditioned continuous control tasks. The key idea is to identify a $\delta$-distilled policy that performs better than the current policy. Then a mixture policy is generated if a $\delta$-distilled policy is identified. Finally, the parameters are updated with behavior cloning to the relabeled data. Experiments on three goal-oriented tasks demonstrate the efficiency of the proposed method.

**Summary Of The Review:**

The method is simple yet effective. However, it heavily relies on the ability of reseting the environment to a previous state, which is unfair to the baselines. The experiments also need improvement.

---

> ### Author Response · Authors · 2021-11-19
> **Response to Reviewer FXhj**
>
> Thank you for your comment. We make the following clarifications to address your concerns about this paper respectively:
>
> **Q1:** *The SELECT requires resetting the environment to a given previous state, which can only be achieved in simulator but not real-world.*
>
> **A1:**  We want to emphasize it is only one aspect of our contributions to design the practical implementation of SPLID, which requires SELECT function. Another major contribution is proposing a novel meta-algorithm SPLID and analyzing its theoretical properties (meta-algorithm: a concept in learning theory, meaning an algorithm learns how to learn). We may instantiate this meta-algorithm to other concrete algorithms by specifying the manners to purse $\delta$-distilled policy.
>
> **Q2:** *The overall learning process is similar to Go-Explore. But Go-Explore is not compared.*
>
> **A2:** Go-explore is not a Deep Reinforcement Learning algorithm, and it has an archive to store all the paths, which requires ample space for storage. Similar to previous papers studying GCRS, [1,2,3], we also do not compare with Go-explore.
>
> **Q3:** *SPLID w/o SELECT does not perform well, which suggests that the method highly relies on the ability to reset the simulator.*
>
> **A3:** On the contrary, It shows the SELECT function is an important component of the practical implementation by phenomenon that SPLID w/o SELECT does not perform well.  Please see Appendix D in the revised paper for more details, which serves as the supplementary explanations of the discussions in Section 4.1.
>
> **Q4:** *It is unclear to me why it is called "meta-algorithm"*.
>
> **A4:** Meta-algorithm is a concept in learning theory, meaning an algorithm learns how to learn.
>
> Based on our proposed meta-algorithm, we can instantiate it to any concrete algorithm which shares the same theoretical properties if we determine the specific protocol to pursue $\delta$-distilled policy and enhance exploration.
>
> **Q5:** *The experiments are weak. The role of $\delta$-distilled policy is unclear.*
>
> **A5**:  As an impractical implementation and Proposition 1, the role of $\delta$-distilled policy is revealed by the importance of SELECT function. Please refer to the third answer (A3) in the response for more details.
>
> **Q6:** *It is unclear how the algorithm improves the quality of "the timesteps were taken to achieve the goal."*
>
> **A6:** We add more discussions in Appendix D and Appendix E in the revised paper, showing that SPLID improves the quality of "the time-steps were taken to achieve the goal."
>
> Here are the reference papers:
>
> [1] Hindsight experience replay, Andrychowicz et al., 2017, arXiv preprint arXiv:1707.01495.
>
> [2] Goal-conditioned imitation learning, Ding et al., 2019, International Conference on Learning Representations 2021.
>
> [3] Policy Continuation with Hindsight Inverse Dynamics, Hao Sun et al., 2019, Advances in Neural Information Processing Systems 2019

---

> > ### Comment · Reviewer_FXhj · 2021-11-23
> > **Thanks for the response**
> >
> > I thank the authors for the response. Some of my questions are answered. However, I am still not convinced of the main concern of using SELECT function. My concern is that the assumption of the ability to resetting environment could be unrealistic. The authors did not directly answer my question.
> >
> > The reason why I propose to compare with Go-Explore is because Go-Explore also assumes the ability to manipulate the environment. I am also not satisfied with authors’ response.
> >
> > Thus, I will maintain my score as reject.

---

### Official Review · Reviewer_Uax1 · 2021-11-02

**Correctness:** 3
**Technical Novelty And Significance:** 3
**Empirical Novelty And Significance:** 2
**Recommendation:** 3
**Confidence:** 3

**Main Review:**

The paper is suffering from a large number of grammatical as well as spelling mistakes that make it very difficult to follow. This goes beyond inelegant use of language and includes puzzling mistakes in technical terms such as “discounted factor” or “total variance”. Based on this alone, the paper is not fit for publication in its current state.

Overall, it is unclear what insight is to be gained from this paper. The proposed concrete implementation is similar to an evolution strategy minimizing the hitting time and as such is broadly sensible; however, it is unclear in which way the proposed method has an advantage over existing approaches such as HER. I find the experimental evaluation to be unconvincing: first, HER can be significantly improved by using a more capable reinforcement learning algorithm and the sample-efficiency gains shown in the plots may be due to this alone. Similarly, it is unclear whether the method is still improving or converging on FetchSlide; final performance on this task should be significantly higher.


**Summary Of The Paper:**

The authors propose an approach for goal-conditioned reinforcement learning via self-imitation. The paper first outlines a meta-algorithm which defines conditions under which an instantiation of the algorithm constitutes a valid policy improvement operator. Following this, the authors propose a concrete instantiation of the algorithm which perturbs the policies parameters, selects policies which decrease the hitting time on randomly chosen goals and then distills the selected behavior into a new policy


**Summary Of The Review:**

The paper is suffering from a large number of grammatical as well as spelling mistakes that make it very difficult to follow. Based on this, the paper is not fit for publication in its current state.

---

> ### Author Response · Authors · 2021-11-19
> **Response to Reviewer Uax1**
>
> Thank you for your review and proposed concerns. Please see below the responses to the concerns you raised.
>
> **Q1:** *The paper is suffering from a large number of grammatical as well as spelling mistakes. This goes beyond inelegant use of language and includes puzzling mistakes in technical terms such as “discounted factor” or “total variance”. *
>
> **A1:** Thank you for your comments. We have fixed the typos and grammar errors.
>
> **Q2:** *HER can be significantly improved by using a more capable reinforcement learning algorithm and the sample-efficiency gains shown in the plots may be due to this alone.*
>
> **A2:** We need to clarify that our proposed method is based on HER technique, while we use the HER implementation in the original paper [1] as the baseline. It is similar to the previous paper [2].
>
> **Q3:** *It is also unclear in which way the proposed method has an advantage over existing approaches such as HER.*
>
> **A3:** As for the reason why the proposed method SPLID has an advantage over GCSL:  SPLID has more high-quality samples to conduct behavior cloning owing to (1) relabel in a broader range (GCSL only relabels the last step) (2) pursue -distilled policy (In our practical implementation, it is equivalent to minimizing First Hit Time). This point has been clarified in the previous paper (Section 4.1). Also, we add more discussions and experiments in Appendix D in the revised paper for your reference. According to previous work on GCSL [2], GCSL improves HER by introducing a supervised learning protocol. Since SPLID improves GCSL by introducing $\delta$-distilled policy and other techniques, it is natural that SPLID also outperforms HER.
>
> Reference papers:
>
> [1] Hindsight experience replay, Andrychowicz et al., 2017, arXiv preprint arXiv:1707.01495.
>
> [2] Goal-conditioned imitation learning, Ding et al., 2019, International Conference on Learning Representations 2021.

---

### Official Review · Reviewer_JnUJ · 2021-11-04

**Correctness:** 4
**Technical Novelty And Significance:** 3
**Empirical Novelty And Significance:** 3
**Recommendation:** 8
**Confidence:** 3

**Main Review:**

- Exposition is clear and the problem is interesting. Authors clearly describe a number of specific concepts necessary to understand exactly what their contribution is, especially as compared to the state of the art.
- Authors give convincing theoretical backing to their approach, as well as conditions under which their key results hold.
- Authors clearly demonstrate an improvement over the state of the art in 3 experiment domains, and compare against several other baselines. Authors also give ablation analysis.

Weaknesses:
- My only major concern is that the state of the art (GCSL) (Ghosh et al. 2019) somehow performs essentially the worst of any baseline in all their experimental domains. The authors state that this is due to the difficulty of the domains -- perhaps then, the authors should include some experimental domains where GCSL performs at least better than trivially poor, so that it does not give the feeling that these set of experimental domains were somehow biased toward the proposed method. Showing the relative performance of the authors' method vs. SOTA in a domain where SOTA can effectively learn would also help more clearly describe exactly where the authors' contribution is having an impact as it relates to the features of different environments or training parameters.


---------------------
Update: I have read the author response. They have adequately addressed my primary concern, so I maintain my score. I recommend that the authors move some of the experiments from appendix E to the main text, since it is more convincing that domains were balanced for the competitor as well as the proposed method.

**Summary Of The Paper:**

The authors consider the Goal-Conditioned continuous control RL task, which is difficult due to reward sparsity. The authors propose a method for training deep RL agents in a more sample efficient manner through self imitation on policy traces that have been "distilled", i.e., randomly perturbed and argmax'd according to some metric like "First Hit Time", then using hindsight experience replay to relabel traces (handling reward sparsity). This improves of over the previous state of the art which did not use the policy distillation step, simply relabeling with the most recent policy, prone to error propagation. The authors demonstrate that their policy distillation step improves the state of the art on 3 RL baseline settings, and also give theoretical backing to their approach and conditions for convergence.


**Summary Of The Review:**

I am rating accept as the writing is clear, motivations and contributions are clear, theoretical results are convincing, and experimental results are mostly good -- the only issue is that the state of the art somehow performs very poorly in the given domains. I hope the authors can either explain this in greater detail or include a domain in the final version where the state of the art is at least able to learn something.

---

> ### Author Response · Authors · 2021-11-19
> **Response to Reviewer JnUJ**
>
> Thank you for your positive comments. Your understanding of our contribution is insightful. Please see the below responses to the concern you raised.
>
> **Q:** *The authors should include some experimental domains where GCSL performs at least better than trivially poor.*
>
> **A:** To address your concern, we add three experiments in Appendix E in the revised paper. In these three environments, GCSL demonstrates non-trivial performance, but our proposed method still achieves better performance.

---

### Decision · Program_Chairs · 2022-01-20

**Decision:**

Reject

**Comment:**

The paper proposes a novel meta-algorithm, called Self-Imitation Policy Learning through Iterative Distillation (SPLID) , which relies on the concept of -distilled policy to iteratively level up the quality of the target data and agent mimics from the relabeled target data.
Several aspects of the paper can be improved. The reviewers are concerned in particular about the experimental section which might not exhaust the core set of tasks, where the method should be compared with baselines. Furthermore the presentation can be significantly improved (lots of grammatical errors). Another major point is the novelty of the presented algorithm.

In the rebuttal the authors tried to address some of the remarks, in particular by adding additional experiments to the empirical section of the paper. Those experiments still do not convince some of the reviewers. Furthermore, one of the biggest concerns is still a limited novelty of the approach. The presentation of the paper still needs to be substantially improved. Thus the paper still requires nontrivial work.